# *APOE2* is associated with longevity independent of Alzheimer's disease

**Mitsuru Shinohara[1,2]\*, Takahisa Kanekiyo[1,3], Masaya Tachibana[1,4], Aishe Kurti[1], Motoko Shinohara[1], Yuan Fu[1], Jing Zhao[1], Xianlin Han[5], Patrick M Sullivan[6], G William Rebeck[7], John D Fryer[1,3], Michael G Heckman[1,8], Guojun Bu[1,3]\***

[1]Department of Neuroscience, Mayo Clinic, Jacksonville, United States; [2]Department of Aging Neurobiology, National Center for Geriatrics and Gerontology, Aichi, Japan; [3]Neuroscience Graduate Program, Mayo Clinic, Jacksonville, United States; [4]United Graduate School of Child Development, Osaka University, Osaka, Japan; [5]Barshop Institute for Longevity and Aging Studies, University of Texas Health Science Center at San Antonio, San Antonio, United States; [6]Duke University School of Medicine, Durham Veterans Health Administration Medical Center's Geriatric Research, Education and Clinical Center, Durham, United States; [7]Department of Neuroscience, Georgetown University Medical Center, Washington, United States; [8]Division of Biomedical Statistics and Informatics, Mayo Clinic, Jacksonville, United States

**Abstract** Although the ε2 allele of apolipoprotein E (*APOE2*) benefits longevity, its mechanism is not understood. The protective effects of the *APOE2* on Alzheimer's disease (AD) risk, particularly through their effects on amyloid or tau accumulation, may confound *APOE2* effects on longevity. Herein, we showed that the association between *APOE2* and longer lifespan persisted irrespective of AD status, including its neuropathology, by analyzing clinical datasets as well as animal models. Notably, *APOE2* was associated with preserved activity during aging, which also associated with lifespan. In animal models, distinct apoE isoform levels, where *APOE2* has the highest, were correlated with activity levels, while some forms of cholesterol and triglycerides were associated with apoE and activity levels. These results indicate that *APOE2* can contribute to longevity independent of AD. Preserved activity would be an early-observable feature of *APOE2*-mediated longevity, where higher levels of apoE2 and its-associated lipid metabolism might be involved.

**\*For correspondence:**
shinohara@ncgg.go.jp (MS);
bu.guojun@mayo.edu (GB)

**Competing interests:** The authors declare that no competing interests exist.

## Introduction

It is well established that the ε4 allele of the apolipoprotein E gene (*APOE4*) is a strong risk factor for late-onset Alzheimer's disease (AD) and that *APOE2* is protective (*Liu et al., 2013*; *Guo et al., 2020*). Additionally, both *APOE* alleles are also associated with longevity. Several case-control studies have shown higher frequencies of *APOE2* in elderly individuals and centenarians compared to younger populations, whereas the frequency of *APOE4* is lower in the older individuals (*Cauley et al., 1993*; *Schächter et al., 1994*; *Sebastiani et al., 2019*). Some longitudinal studies have also demonstrated beneficial effects of *APOE2* and deleterious effects of *APOE4* on longevity (*Corder et al., 1996*; *Rosvall et al., 2009*). Effects of *APOE* on longevity have also been observed in unbiased GWAS analyses (*Deelen et al., 2011*; *Nebel et al., 2011*). Despite the accumulated evidence regarding *APOE* effects on longevity, the underlying mechanism has rarely been addressed.

ApoE proteins form lipoprotein particles and regulates lipid transport in both the central nervous systems and periphery. While apoE isoforms can directly affect Alzheimer's neuropathology, including the accumulation of amyloid-β (Aβ) and tau, increasing evidence has demonstrated that apoE

isoforms also contribute to cognitive function through AD neuropathology-independent pathways (*Liu et al., 2013*; *M. Di Battista et al., 2016*; *Montagne et al., 2020*; *Yamazaki et al., 2020*). These effects might be mediated by apoE-regulated lipid metabolism, synaptic function, vascular integrity, and/or neuroinflammation (*Liu et al., 2013*; *M. Di Battista et al., 2016*; *Montagne et al., 2020*; *Yamazaki et al., 2020*; *Williams et al., 2020*; *Najm et al., 2019*). Recently, by performing both clinical and preclinical analysis, we showed that *APOE2* protects against age-associated cognitive decline independent of AD neuropathology. Interestingly, these effects were also independent of synaptic and neuroinflammatory changes but associated with cholesterol metabolism in both the brain and periphery (*Shinohara et al., 2016*).

In this study, we aimed to address the effects of *APOE* on longevity and assess their relationship with AD, including clinical symptoms and neuropathology. Toward this, we analyzed clinical records of a large number of subjects enrolled by the National Alzheimer's Coordinating Center (NACC). In addition, we analyzed mouse models expressing each human apoE isoform under the control of the mouse endogenous promoter (targeted replacement or 'TR' mice). Our results provide important insights into how *APOE2* contributes to longevity.

## Results

### *APOE* is associated with longevity independent of AD in clinical cohorts

To evaluate relationships of each *APOE* allele with longevity, we analyzed NACC clinical records that collected clinical and neuropathology data of large number of demented and non-demented people in a longitudinal manner (*Weintraub, 2009*). In NACC subjects whose *APOE* genotype, sex, and race were available (n = 24,661, demographics shown in *Supplementary file 1a*), 5413 (21.9%) of subjects were recorded as dead. We analyzed their survival after birth. As shown in *Figure 1A*, compared to subjects with ε3/ε3 genotype (i.e. '*APOE3*'), those who were ε3/ε4 or ε4/ε4 (i.e. '*APOE4*') had worse survival, while individuals who had ε2/ε2 or ε2/ε3 genotype (i.e. '*APOE2*') had better survival. Notably, these effects were persisted in analyses that adjusted for sex, race, AD, cognitive

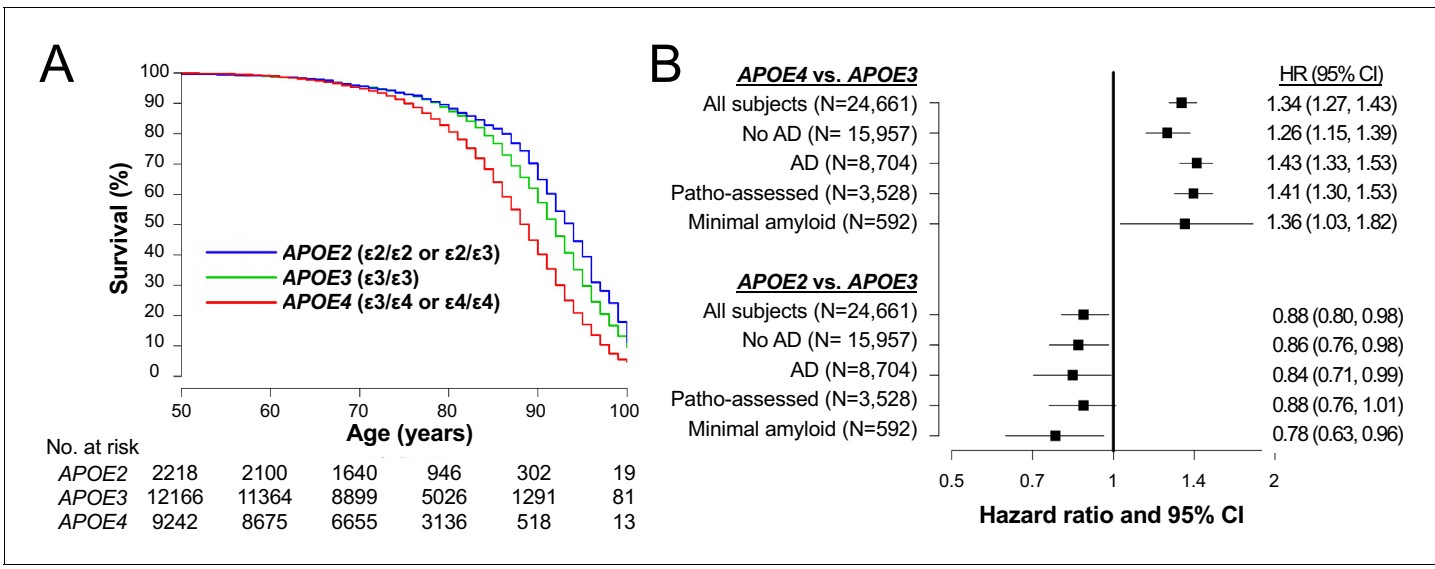

**Figure 1.** *APOE* associate with lifespan irrespective of AD in clinical cohorts. (**A**) Kaplan-Meier survival curve according to *APOE* genotype. (**B**) Effects of *APOE4* (ε3/ε4 or ε4/ε4) or *APOE2* (ε2/ε3 or ε2/ε2) on survival compared to *APOE3* (ε3/ε3) as a reference in all subjects ('All subjects'), when stratifying by AD diagnosis at last visit ('No AD' or 'AD'), in subjects with neuropathologically-assessment ('Patho-assessed'), and subjects with minimal amyloid pathology ('Minimal amyloid'). HR = hazard ratio. CI = confidence interval. HRs and 95% CIs result from Cox proportional hazards regression models. Models for all subjects were adjusted for sex, race, cognitive status at last visit, presence of AD at last visit, and cardiovascular factors. The models for subjects with neuropathological assessment were adjusted for sex, race, CERAD diffuse plaque score, CERAD neuritic plaque score, Braak NFT stage, presence of vascular pathology, and cardiovascular factors. All other models were adjusted for sex, race, and cardiovascular factors.

status, and cardiovascular factors in all subjects ('All subjects' in *Figure 1B*). Moreover, stratifying non-AD and AD status keep these *APOE* effects significant ('No AD' and 'AD' in *Figure 1B*).

We subsequently focused only on the subgroup of 3528 subjects who were neuropathologically assessed after their death (demographics shown in *Supplementary file 1b*) to evaluate the relationships between *APOE* and longevity while accounting for AD-related neuropathology. When adjusting for sex, race, diffuse plaque score, neuritic plaque score, NFT stage, the vascular pathology, and cardiovascular factors, survival was again worse for *APOE4* subjects and better for *APOE2* subjects compared to the *APOE3* group ('Patho-assessed' in *Figure 1B*). Moreover, the harmful effects of *APOE4* and the beneficial effects of *APOE2* on mortality were even observed in individuals without remarkable amyloid pathology ('Minimal amyloid' in *Figure 1B*).

These results were precisely described in *Supplementary file 1c*. Although not of primary interest, ε2/ε4 genotype was mildly (but not significantly) associated with worse survival compared to ε3/ε3 (*Supplementary file 1c*). These results indicate that *APOE* contributes to longevity independent of clinical and neuropathological status of AD. However, as in any epidemiologic study, unmeasured confounding variables could have affected our results.

## *APOE2* benefits lifespan in animal models, associated with preserved activity

We thus analyzed human apoE-targeted replacement (apoE-TR) (*Sullivan et al., 1997*; *Sullivan et al., 1998*; *Knouff et al., 1999*) and *Apoe*-knockout (KO) mice (*Piedrahita et al., 1992*) to examine whether the effects of *APOE* on lifespan can indeed be observed in the absence of AD. In the absence of overexpressing mutant *APP* or mutant *MAPT* gene, these mice do not display remarkable amyloid or tau deposition such as those seen in AD (*Shinohara et al., 2016*; *Wang et al., 2005*; *DiBattista et al., 2016*). In the survival mouse cohort (n = 118), due to the regulation of our animal experiment, we euthanized mice (33.9% of total animals) when they showed severe age-related symptoms including severe weakness, masses, significant skin lesions etc. that were diagnosed by the expert on-site veterinarian to avoid severe pain and distress before death (see Materials and methods), while there was no difference in *APOE* effects on these severe symptoms (*Supplementary file 1d*). The remaining 66.1% of mice died naturally. During their lives, we performed a non-invasive battery of behavioral tests including the open-field assay (OFA) and elevated plus maze (EPM) at both young (4–7 months age) and old (21–24 months age) ages, and rotarod test at old age (*Figure 2A*). Overall, as shown in *Figure 2B*, apoE2-TR mice showed a reduced risk of all-cause mortality when compared to apoE3-TR (p=0.0004), apoE4-TR (p=0.007), and *Apoe*-KO (p=0.0001). Notably, as the survival pattern for 'control' apoE3-TR mice varied over time, we also made pair-wise comparisons of 'early survival' with the apoE3-TR group by censoring all survival times at day 800. For these comparisons of 'early survival', we observed worse survival for apoE4-TR (p=0.013). Additionally, the median age at death was in the order of apoE2-TR (911 days) >apoE3 TR (825 days)>apoE4 TR (753 days)>*Apoe* KO (738 days), where apoE4-TR mice showed ~10% decrease in median lifespan, compared to apoE3-TR mice (*Figure 2B*). Nonetheless, these animal model data support beneficial effects of *APOE2* on mortality in the absence of AD.

In the OFA (*Figure 2C*) performed at old age, apoE2-TR mice generally showed greater locomotor and exploratory activity, compared to other mice (*Figure 2D–F*). In particular, number of rearing events was significantly increased in apoE2-TR mice even when compared to 'control' apoE3-TR mice (*Figure 2F*). Notably, the effect of *APOE* genotype on these activity levels was more prominent for the number of rearing events (p<0.0001), which better reflected the exploratory behavior, than the distance traveled (p=0.0005) or the time spent mobile (p=0.0056). Moreover, when comparing the scores at old age with the ones performed at young age, we observed that apoE2-TR mice showed less of an age-associated decrease in these activities (*Figure 2G&H*). More interesting, significant correlations were observed between the activity levels in the OFA and age at death across *APOE* genotypes (*Figure 2I–K*). There were no significant changes in their body weights of this cohort (data not shown). We also observed some trends of positive association (though not significant, p=0.059) between age at death and results of EPM test (*Figure 2L&M*), which primarily measure the exploratory/anxiety phenotype, though no correlations for rotarod test that purely measure locomotive ability (*Figure 2N&O*).

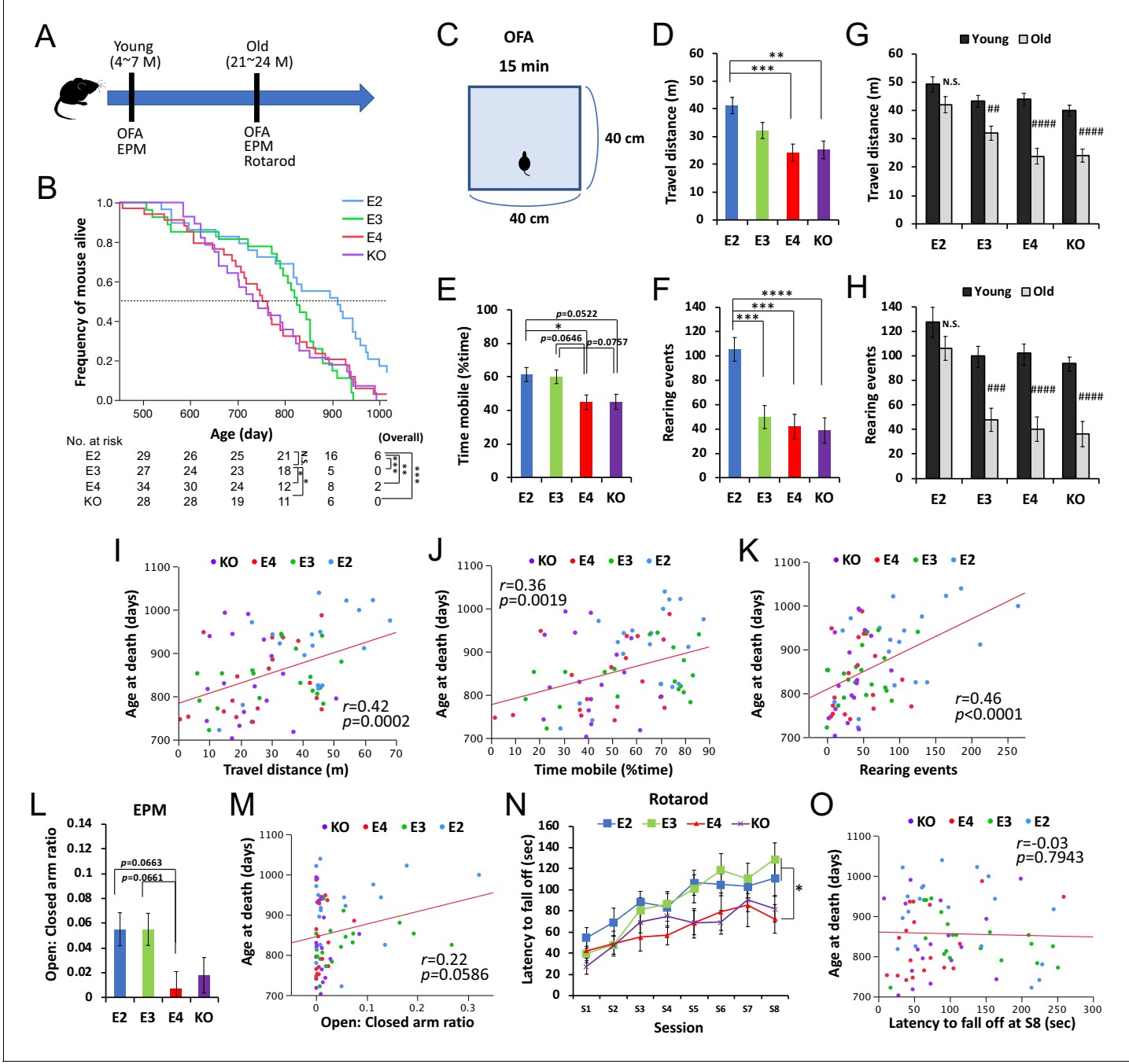

**Figure 2.** *APOE2* benefits lifespan and preserved activity levels in animal models. (**A**) The timeline of the survival cohort. The OFA and EPM were performed at both young (4–7 months) and old (21–24 months) ages. Rotarod test was also performed at old age. (**B**) Kaplan-Meier survival curve of apoE-TR and *Apoe*-KO mouse models (n = 118), categorized by *APOE* carrier status with number of mice at risk. *p<0.05, **p<0.01, ***p<0.001 after Bonferroni correction for multiple comparisons; calculated by Cox proportional hazards regression models that were adjusted for sex. Median survival for each group is the point at which the Kaplan-Meier survival curves intersects with the dotted horizontal line. (**C**) Diagram of OFA. (**D–I**) Total distance traveled (D and G), percent time spent mobile (**E**), and number of rearing events (F and H) in the OFA were compared among *APOE* genotype groups at old age (**D–F**) or between young and old ages within the same *APOE* genotype (G and H) after adjusting for sex (n = 17–20 mice/group). (**I–J**) Total distance traveled (I), percent time spent mobile (**J**), and number of rearing events (**K**) in the OFA performed at old age are plotted against the age at death. (L and M) Ratios of time stayed at open arm vs. closed arm measured by EPM test at old age were compared among *APOE* genotype groups after adjusting for sex (**L**), and plotted against following age at death (**M**) (n = 17–20 mice/group). (N and O) Latency to fall off at each session of rotarod test at old age was compared among *APOE* genotype groups after adjusting for sex (**N**), and plotted against following age at death (**O**) (n = 17–20 mice/group). Data are presented as adjusted means ± standard errors of the means. *p<0.05, **p<0.01, ***p<0.001, ****p<0.0001;

*Figure 2 continued on next page*

*Figure 2 continued*

compared among *APOE* genotype groups using the Tukey's HSD test (D-F and L) or repeated-measures one-way ANOVA followed by Tukey-Kramer test (N)., or #p<0.05, ##p<0.01, ###p<0.001, ####p<0.0001; compared with young mice using two-sided Student's t test (G and H). (I-K, M and O) Correlation coefficients (r) and p-values were calculated using the Pearson correlation test. N.S. = not significant.

The online version of this article includes the following source data for figure 2:

**Source data 1.** Survival data of apoE-TR and *Apoe*-KO mouse for *Figure 2A* as well as *Supplementary file 1d*.
**Source data 2.** Scores in the OFA and EPM and age at death for *Figure 2D–F & I–M*.
**Source data 3.** OFA scores between young and old for *Figure 2G&H*.
**Source data 4.** Rotarod scores and age at death for *Figure 2N&O*.

## *APOE2* is associated with preserved activity in clinical cohorts

These results prompted us to re-review the NACC records to examine the clinical relevance of *APOE* effects on activity. As a candidate variable that might reflect the results observed in the animal experiments, we searched for 'activity' in NACC variables, and identified 'dropped activities and interests' as one component of the geriatric depression scale (GDS) test. Despite its subjective nature, previous studies suggest that this question could reflect age-related changes of some form of physical activity independent of depression (*Scheetz et al., 2012*; *Rabaglietti et al., 2011*). As these GDS components could be confounded by cognitive status, we analyzed subjects who were cognitively normal at their final GDS evaluation. We also assessed only subjects who were over 60 years old in order to assess age-related changes of this measure, resulting in a sample size of 9530 subjects. Compared to *APOE3* subjects, we observed a lower frequency of 'dropped activities in interests' for *APOE2* subjects (Odds ratio [OR]=0.80, p=0.019) in analysis that was adjusted for sex, race, and age, while there was no significant difference between *APOE3* and *APOE4* subjects (OR = 0.98, p=0.82) (*Figure 3A*, top). Subsequently, comparisons of the remaining 14 GDS items as well as the GDS score were also made between the three *APOE* groups in order to evaluate whether *APOE* genotype is systematically associated with depression. We observed no such evidence; total

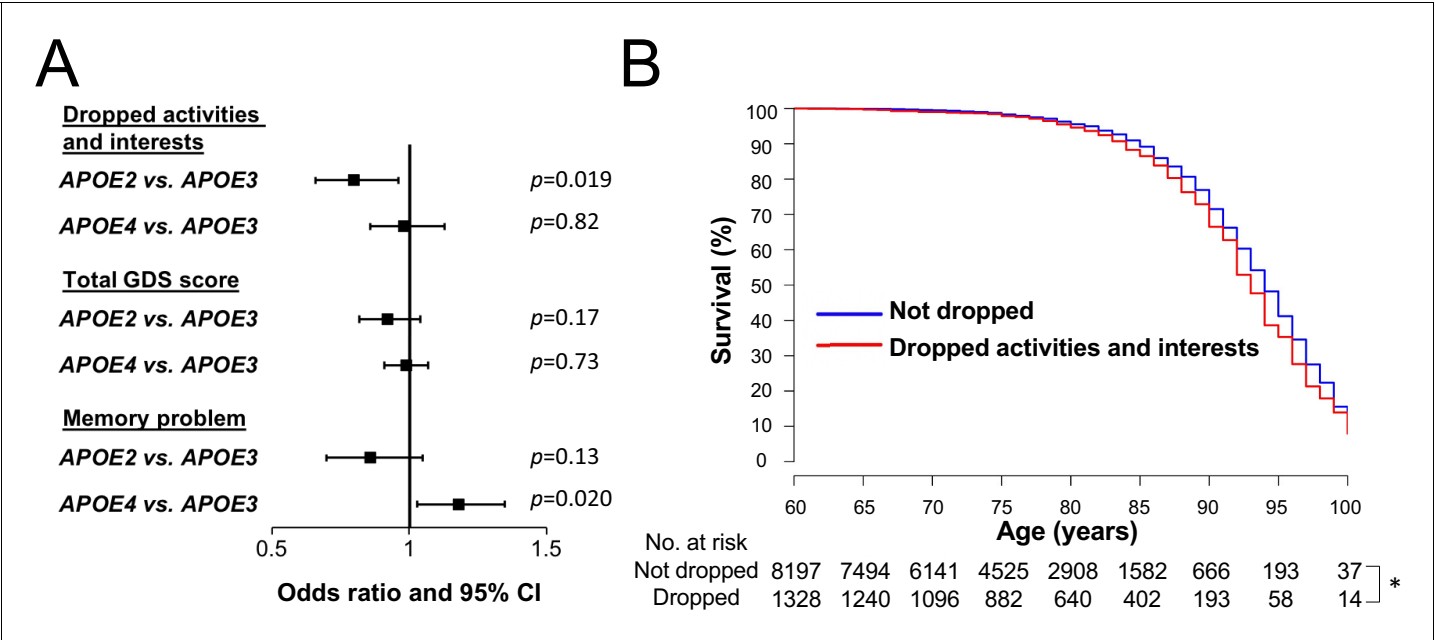

**Figure 3.** *APOE2* is associated with preserved activity in clinical cohorts. (**A**) Odds ratio with 95% CIs of *APOE2* or *APOE4* on 'dropped activities and interests', 'total GDS score', and 'memory problem' compared to *APOE3*, as calculated by logistic regression models that were adjusted for sex, race, and age at the time of the GDS questionnaire. (**B**) Kaplan–Meier survival curve of subjects with/without 'dropped activities and interests'; dropped activities and interests were associated with poorer survival (Hazard ratio = 1.20, p=0.010). *p<0.05; calculated by Cox proportional hazards regression analysis adjusting for sex, race, and *APOE* genotype group.

GDS score was not significantly different in comparison to *APOE3* subjects for either *APOE2* subjects or *APOE4* subjects (*Figure 3A*, middle), and 'feeling helpless' was the only other individual GDS item that differed significantly between *APOE2* and *APOE3* subjects (OR = 0.69, p=0.005) (*Supplementary file 1e*). Notably, consistent with previous findings (*Shinohara et al., 2016*; *Hyman et al., 1996*; *Wilson, 2002*), the risk of 'problems with memory' was significantly higher for *APOE4* compared to *APOE3* subjects (OR = 1.18, p=0.020) (*Figure 3A*, bottom), supporting the validity of our other GDS findings. Moreover, in exploratory analysis using 'dropped activities and interest' as a marker of lifetime activity level in this same cohort of 9530 subjects, we observed that 'dropped activities and interests' was associated with a shorter lifespan in analysis adjusted for sex, race, and *APOE* genotype (HR = 1.20, p=0.010) (*Figure 3B*). These clinical data are consistent with the animal data indicating that some form of activity in the lifetime is associated with *APOE2*-mediated longevity.

## Activity associates with apoE and lipid levels in animal models

We then analyzed biochemical changes in the CNS and periphery of an independent mouse cohort that was harvested within one month of the OFA test (*Figure 4A*, young group = 5–7 months of age, old group = 21–24 months of age, *Supplementary file 1d*). Consistent with the survival cohort, old apoE2-TR mice had a greater ability to retain their activity levels in the OFA (*Figure 4B–D*). The relative levels of apoE in the brain, CSF and plasma were in the order of apoE2-TR > apoE3 TR>apoE4 TR, as previously reported (*Shinohara et al., 2016*). Activity levels in the OFA correlated with such different apoE levels in these tissues of old mice across *APOE* genotypes (*Figure 4E–H* and *Supplementary file 1f*). We analyzed changes in synaptic or inflammatory markers, including PSD95, GFAP, CD11b, IL1β, TNFα, and CRP in the brain or plasma (*Shinohara et al., 2016*), but did not observe any significant differences that were associated with the activity levels in the OFA (*Supplementary file 1f*). Brain cholesterol levels, which were measured by lipidomic analysis (*Shinohara et al., 2016*), were lower in apoE2-TR mice, and higher in *Apoe*-KO mice, compared to apoE3-TR or apoE4-TR mice (*Figure 4I*). Interestingly, brain cholesterol levels were inversely correlated with apoE levels in the brain (*Figure 4J*) and the activity levels in the OFA (*Figure 4K*). Among plasma cholesterols, only HDL-cholesterol levels, which were lowest in *Apoe*-KO mice (*Figure 4L*) were associated with the activity levels in the OFA across *APOE* genotypes (*Figure 4M* and *Supplementary file 1f*). Notably, plasma triglyceride levels, which were highest in apoE2-TR mice (*Figure 4N*), were also associated with the activity levels in the OFA (*Figure 4O* and *Supplementary file 1f*) and plasma apoE levels (*Figure 4P*). These plasma lipid profiles showed trends toward age-associated decreases in HDL-cholesterol levels in *Apoe*-KO mice, and triglyceride levels in apoE4-TR and apoE3-TR mice. Notably, those age-associated decreases were not observed in apoE2-TR mice (*Figure 4Q&R*). These animal data suggest that distinct apoE isoform levels, where *APOE2* has the highest, and associated lipid metabolism, might be involved in the beneficial effects of *APOE2* on longevity phenotypes.

## Discussion

Despite strong evidence that *APOE2* contributes to longevity, the underlying mechanism is not clear. Because *APOE2* and *APOE4*, respectively, reduces and increases the risk of AD, there is a critical need to address whether the effects of *APOE* genotype on longevity are mediated through such AD risks. In this study, we uniquely combined human and mouse model studies and observed that *APOE2* can contribute to longevity independent of AD risks.

Although many clinical studies have observed that *APOE* is associated with longevity, no studies have succeeded in fully distinguishing the longevity effects of *APOE* from its AD effects. NACC database contains both clinical and neuropathological data of large number of AD patients as well as cognitively normal subjects with *APOE* genotypes. Such peculiarities of NACC database indeed allowed us to evaluate whether *APOE*-associated longevity is related to AD or not, including its neuropathology. While we observed significant *APOE* effects on mortality by adjusting cognitive status, or by stratifying AD and non-AD status, we did not observe significant *APOE* effects in cognitively normal individuals, although there are the same trends of difference (data not shown). One possible reason is that compared to demented subjects, there is a smaller number of death event in cognitively normal individuals in NACC database, making it difficult to assess their survival. Another

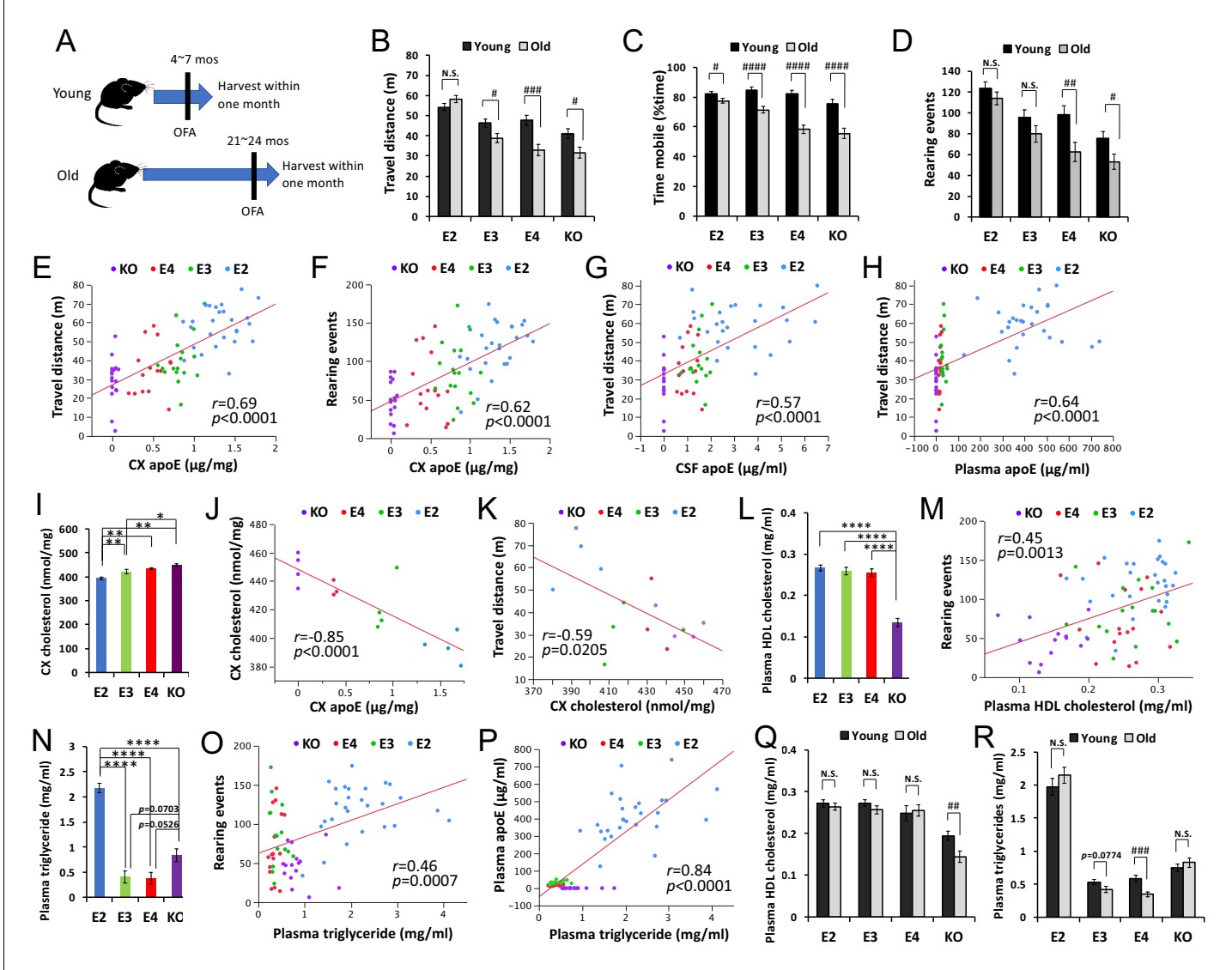

**Figure 4.** ApoE and lipid levels correlate with activity. (A) The timeline of biochemically assessed cohort. The OFA was performed within one month before harvest. (B–D) Total distance traveled (B), percent time spent mobile (C), and number of rearing events (D) were compared between young and old mice within the same *APOE* genotype after adjusting for sex (n = 16–28 mice/group). (E–H) ApoE levels in the brain cortex (CX, E, and F), CSF (G) and plasma (H) are plotted against the total distance traveled (E, G and H), or number of rearing events in the OFA (F). (I) Levels of cholesterol in the brain cortex of old male mice were compared among *APOE* genotypes (n = 3–4 mice/group) and are plotted against apoE levels in the cortex (J) and total distance traveled in the OFA (K). (L–R) Plasma HDL-cholesterol levels (L and M) or triglyceride levels (N–P) were compared among *APOE* genotypes in old mice (L and N) or between young and old mice within the same *APOE* genotype (Q and R) after adjusting for sex and plotted against the number of rearing events in the OFA (M and O) or plasma apoE levels (P) (n = 9–27 mice/group). Data are presented as adjusted means ± standard errors of the means. *p<0.05, **p<0.01, ***p<0.001, ****p<0.0001; compared among *APOE* genotypes using the Tukey-Kramer test (I), Tukey's HSD test (L and N), or #p<0.05, ##p<0.01, ###p<0.001, ####p<0.0001; compared between young and old mice using two-sided Student's t test (B-D, Q, and R). (E-H, J, K, M, O, and P) Correlation coefficients (r) and p-values were calculated using the Pearson correlation test. N.S. = not significant.

The online version of this article includes the following source data for figure 4:

**Source data 1.** OFA scores between young and old for *Figure 4B–D*.

**Source data 2.** OFA scores and several biochemical measurements for *Figure 4E–H and L–P* as well as *Supplementary file 1f*.

**Source data 3.** Brain cholesterol and OFA scores for *Figure 4I–K*.

**Source data 4.** Plasma lipids between young and old for *Figure 4Q and R*.

possible reason is that as *APOE2* can also protect against age-related cognitive decline independent of AD (*Shinohara et al., 2016*), cognitive decline could be a downstream event of *APOE*-associated pathways related to longevity independent of AD. Longer follow-up would be necessary to address such possibilities.

To strengthen our conclusion regarding NACC clinical data analyses, we analyzed animal models without AD pathology to address human *APOE2* effects on longevity. While we observed significant beneficial effects of *APOE2* on lifespan, compared to *APOE3*, *APOE4* and *Apoe*-KO, we did not observe a significant difference between *APOE4* versus *APOE3* in the overall survival period (*Figure 2B*). Although we observed some harmful effects of *APOE4* on lifespan, as observed in 'the early survival' and median age at death, which many previous longevity studies primarily evaluate (*Ladiges et al., 2009*), further studies would be needed to confirm that *APOE4* has indeed worse impacts on overall lifespan, compared to *APOE3*, as observed in clinical data. Nonetheless, this is the first study showing the effect of *APOE*, especially for the beneficial effects of *APOE2*, on longevity in animal models in the absence of amyloid or tau pathology found in AD.

While we obtained evidence that preserved activity is associated with *APOE2*-related longer lifespan from both clinical and animal studies, similar phenomena were indeed observed in other models of longevity. Calorie restriction, the most validated method to promote longevity in various organisms, was observed to enhance or preserve physical activity, spontaneous locomotive activity or voluntary wheel running activity (*Yu et al., 1985*; *Weed et al., 1997*; *Bross et al., 2005*; *Katewa et al., 2012*). Notably, *Ingram et al., 1982* observed in aged C57BL/6J mice, while exploratory activity was associated with lifespan, similar to our findings, other forms of physical activity, such as the scores of rotarod tests, were not, consistent with this study. Although it is not clear whether such kind of activity is directly involved in prolonged lifespan (*McCarter et al., 1997*), it would be a potential surrogate maker reflecting longer lifespan. Thus, it should be further investigated how longevity-associated preserved activity can be objectively evaluated in humans in addition to the subjective GDS questionnaires used in the current study.

Significant associations between the levels of activity and those of apoE protein and related lipids (*Figure 4*) would provide important insights into *APOE* effects on longevity. However, the significant correlations were observed mostly by analyzing groups across *APOE* genotypes, not within each *APOE* genotype. Indeed, several positive correlations between activity levels and apoE or lipid levels disappeared after stratifying *APOE* genotype, while correlations between apoE and lipid levels mostly survived even after stratifying *APOE* genotype (data not shown). One possibility is that animal behavior scores due to individual variations (*Shinohara et al., 2016*; *Lathe, 2004*), or the time lag (~1 month) between behavior test and tissue harvesting could contribute additional variability to observe clear correlation within each genotype with limited number of mice. Another possibility is that factors other than apoE and associated lipids might be involved in *APOE2*-mediated longevity. Nonetheless, previous studies have shown that higher HDL-cholesterol levels in plasma are associated with longer lifespan, although effects of plasma total cholesterols were contradictory (*Weverling-Rijnsburger et al., 1997*; *Upmeier et al., 2009*; *Barzilai et al., 2001*; *Milman et al., 2013*). Moreover, recent studies demonstrate that triglycerides can promote longevity in flies and yeasts (*Katewa et al., 2012*; *Handee et al., 2016*). These studies also observed that triglyceride levels are associated with more activity (*Katewa et al., 2012*). Thus, higher HDL-cholesterol and triglyceride levels regulated by apoE2 might both contribute to *APOE2*-mediated longevity although further study is necessary to confirm such findings and address the potential underlying mechanisms.

There are potential limitations of our study; regarding clinical data analysis, (1) NACC is not a population-based study, but a referral- or volunteer-based study, making it inappropriate to estimate the life expectancy of general population, (2) in NACC database, despite a longitudinal design, many participants were followed in a short period, (3) cause of death are unavailable, and (4) we did not consider all potential confounding factors associated with death, such as cancer, and impairment of vascular function (though we have adjusted for vascular pathology, and cardiovascular factors). Indeed, recent studies reported that breakdown of blood–brain barrier (BBB) plays key roles in *APOE4*-mediated cognitive decline, independent of AD pathology (*Montagne et al., 2020*; *Nation et al., 2019*), which can be protected by *APOE2* (*Conejero-Goldberg et al., 2014*). The possibility of involvement of such factors in *APOE2*-associated longevity should be addressed in future studies. Regarding a major limitation in animal experiments, we have analyzed ε2/ε2 homozygous mice that categorically develop hyperlipidemia, while only a small percentage of human ε2/ε2

carriers develop hyperlipidemia (*Sullivan et al., 1998*; *Mahley et al., 1999*). Though we observed such hyperlipidemia phenotype (i.e., high triglyceride levels) associated with activity levels, further investigation of *APOE2* effects with/without hyperlipidemia (e.g., by using ε2/ε3 animal models) should provide additional insights into how *APOE2* contributes to longevity.

In summary, this study provides complementary evidence from clinical and preclinical data that *APOE2* promotes longevity independent of AD. Our results also indicate that preserved activity during aging might be a good surrogate marker underlying *APOE2*-associated longevity. Moreover, our animal experiments suggest that apoE protein levels, where *APOE2* shows the highest, and associated lipid metabolism might play important roles in *APOE2*-associated longevity. As *APOE2* is a leading longevity gene, understanding the related mechanisms should provide critical insights into healthy living and longevity in our aging society.

# Materials and methods

## Key resources table

| Reagent type (species) or resource | Designation | Source or reference | Identifiers | Additional information |
|---|---|---|---|---|
| Genetic reagent (*Mus. musculus*) | *Apoe*$^{tm1(APOE*2)Mae}$ or apoE2-TR | *Sullivan et al., 1998* | RRID:MGI:3695702 | |
| Genetic reagent (*Mus. musculus*) | *Apoe*$^{tm2(APOE*3)Mae}$ or apoE3-TR | *Sullivan et al., 1997* | RRID:MGI:3695698 | |
| Genetic reagent (*Mus. musculus*) | *Apoe*$^{tm3(APOE*4)Mae}$ or apoE4-TR | *Knouff et al., 1999* | RRID:MGI:4355228 | |
| Genetic reagent (*Mus. musculus*) | *Apoe*$^{tm1Unc}$ or *Apoe*-KO | *Piedrahita et al., 1992* | RRID:MGI:4358709 | |
| Antibody | Goat polyclonal anti-ApoE-biotin | Meridian Life Science | Cat#: K74180B | ELISA detection (1:4000) |
| Antibody | Goat polyclonal anti-Apolipoprotein E Antibody | Millipore | Cat#: AB947 | ELISA capture (1:4000) |
| Peptide, recombinant protein | ApoE3 | Fitzgerald | Cat#: 30 R-2381 | ELISA standard |
| Commercial assay or kit | Cholesterol E | Wako | Cat. #: 999–02601 | |
| Commercial assay or kit | L-Type Triglyceride M | Wako | Cat. #: 994–02891 | |
| Commercial assay or kit | HDL-Cholesterol E | Wako | Cat. #: 997–01301 | |
| Software, algorithm | SAS | SAS Institute, Inc | version 9.4 | |
| Software, algorithm | R Statistical Software | R Foundation for Statistical Computing | version 3.2.3 | |
| Software, algorithm | JMP Pro | SAS Institute, Inc | version 12 | |
| Software, algorithm | AnyMaze software | Stoelting Co | | Animal behavior test |

## Human clinical and neuropathological data

The clinical data from NACC, which were collected by the 34 past and present Alzheimer's Disease Centers (ADCs) from September 2005 to November 2016 as the longitudinal Uniform Data Set (*Weintraub, 2009*), were assessed in this study. We restricted our analysis to 24,661 subjects whose sex, race, and *APOE* genotypes were available in order to assess the effects of *APOE* on longevity (~70% of total subjects). NACC subjects are regarded as a referral-based or volunteer series, and the majority of these subjects are Caucasians (84.2%) with variable cognitive status at final visit (36.0% normal cognition, 4.2% impaired but not MCI, 15.6% MCI, 44.3% dementia) (*Supplementary file 1a*). This cumulative database also contains neuropathological data from a

subset of deceased individuals (N = 3,528, 65.2% of total deceased subjects). For comparison with the *APOE* ε3/ε3 genotype (i.e. *APOE3*) in the primary analysis, *APOE* ε2/ε2 and ε2/ε3 genotypes were grouped together into an *APOE2* group, while *APOE* ε3/ε4 and ε4/ε4 genotypes were grouped together into an *APOE4* group. Although not the primary focus of this study, comparisons between *APOE* ε2/ε4 and ε3/ε3 genotypes were also made, as shown in *Supplementary file 1*.

NACC demographic variables that were utilized in our study included death, age at death, age at initial visit, age at final visit, sex, and race. Degree of cognitive impairment was assessed by cognitive status at final visit using the NACCUDSD variable (which describes cognitive status: normal cognition, impaired but not mild cognitive MCI, MCI, or dementia), and also by clinically diagnosed AD at final visit. Clinically diagnosed AD at final visit was defined as presence of both dementia on NACCUDSD, and a value of 'yes' on NACCALZD (which describes etiologic diagnosis of AD). Cardiovascular factors (occurrence of these at any visit was assessed) that were measured included hypertension, transient ischemic attack, pacemaker, angioplasty/endarterectomy/stent, heart attack/cardiac arrest, cardiac bypass procedure, atrial fibrillation, hypercholesterolemia, congestive heart failure, and stroke. We also utilized data from the geriatric depression scale (GDS), where answers to 15 different yes/no questions are recorded as well as a total GDS score that ranges from 0 to 15 and represents the sum of the scores for the individual GDS questions.

Neuropathological information was collected regarding Consortium to Establish a Registry for Alzheimer's Disease (CERAD) diffuse plaque score, CERAD neuritic plaque score, Braak NFT stage, and the presence of vascular pathology. It is of note that the neuropathologically-assessed cohort had a different frequency of dementia and AD, compared to the overall cohort, with much higher rates of dementia (78.3 vs. 44.3%) and AD (58.0 vs. 35.3%) at final NACC visit, likely due to higher rate of postmortem analysis of demented subjects (*Supplementary file 1b*). Minimal amyloid pathology was defined as both no neuritic plaques and no or sparse diffuse plaques (*Shinohara et al., 2016*).

## Statistical analysis of NACC human clinical data

Statistical analyses were performed using SAS (version 9.4; SAS Institute, Inc, Cary, North Carolina) and R Statistical Software (version 3.2.3; R Foundation for Statistical Computing, Vienna, Austria). The Kaplan-Meier method was used to estimate survival (i.e. after birth), where censoring occurred at the date of the last NACC visit for subjects who did not die. In the primary analysis including all subjects, Cox proportional hazards regression models were used to compare survival between *APOE* carriers. Models were first adjusted for sex and race, and then were subsequently adjusted for cognitive status at final NACC visit and clinically diagnosed AD at final NACC visit in order to account for these two strong confounding factors that are related to both *APOE* genotype and survival. We also stratified by the AD/non-AD status and compared survival using Cox regression models that were adjusted for sex and race. In a sensitivity analysis, we additionally adjusted our Cox models for the aforementioned 10 cardiovascular factors in the subset (97.7%) of subjects who had this information available.

We subsequently examined only neuropathologically assessed cases in order to evaluate the relationship between *APOE*-mediated longevity and AD-related neuropathology. Specifically, Cox proportional hazards regression models that were adjusted for sex, race, CERAD diffuse plaque score, CERAD neuritic plaque score, Braak NFT stage, and the presence of vascular pathology were utilized to compare survival between *APOE* carriers. These comparisons were also made separately in subjects with minimal amyloid pathology, where Cox proportional hazards regression models were adjusted for sex and race. Finally, additional adjustment for cardiovascular factors in Cox regression models was examined in a sensitivity analysis.

In order to adjust for the fact that two primary comparisons of survival were made (i.e. between *APOE3* and *APOE4* subjects, and also between *APOE3* and *APOE2* subjects), we applied a Bonferroni correction after multiple testing, after which p-values<0.025 were considered as statistically significant. All statistical tests were two-sided.

For analysis of the GDS individual items and total score, we restricted the analysis to subjects who were cognitively normal at their last evaluation in order to exclude the possibility of obtaining false answers from cognitively impaired subjects. Comparisons between *APOE3* and *APOE4* subjects, and also between *APOE3* and *APOE2* subjects, were made using binary logistic regression models for the 15 individual yes/no items, and using proportional odds logistic regression models

for the ordinal 0–15 total GDS score (where total GDS scores $\geq$ 10 were collapsed together into one category due to their low frequencies). Models were adjusted for sex, race, and age at the time of the GDS questionnaire. Odds ratios (ORs) and 95% CIs were estimated and are interpreted as the multiplicative increase in the odds of presence of the given characteristic for analysis of individual GDS items, and as the multiplicative increase in the odds of a higher score for analysis of the total GDS score. The primary comparison was regarding the GDS item 'have you dropped many of your activities and interests'. In secondary analysis, comparisons of the remaining GDS items as well as the GDS score were also made to examine whether *APOE* genotype is systematically associated with depression rather than with the more specific 'dropped activities and interests' GDS question. In order to adjust for the two comparisons of each GDS item, we applied a Bonferroni correction separately for each GDS outcome measure, after which p-values<0.025 were considered as statistically significant. Additionally, we evaluated the association between the GDS 'dropped activities and interests' item and survival using a Cox proportional hazards regression model that was adjusted for sex, race, and *APOE* genotype group, where the same group of subjects who were cognitively normal and older than 60 at the time of GDS evaluation were examined. Again, censoring occurred at the date of the last NACC visit for subjects who did not die. All statistical tests were two-sided.

## Animals

Human apoE-TR mice expressing human apoE2, apoE3, or apoE4, under the control of the mouse apoE promoter, known as apoE-targeted replacement (apoE-TR) (*Sullivan et al., 1997*; *Sullivan et al., 1998*; *Knouff et al., 1999*), and *Apoe*-knockout (KO) mice (*Piedrahita et al., 1992*) on a pure C57BL/6 background were used in this study. All cohorts examined in this study were generated from homozygous breeding pairs, group housed without enrichment structures in a specific pathogen-free environment in ventilated cages and used in experiments according to the standards established by the Mayo Clinic Institutional Animal Care and Use Committee (IACUC, Protocol# A58312). We analyzed two animal cohorts in this study (*Supplementary file 1d*). Sample size was determined according to the previous publications related to mouse longevity study (*Ladiges et al., 2009*) as well as the capacity of our animal facility. The first cohort was employed for the survival analysis, and the second was employed for biochemical analyses. In the first cohort used for the survival analysis (n = 118), we euthanized mice when they showed severe age-related symptoms (33.9% of total animals) including severe weakness (a severely hunched posture, rapid weight loss (10% decrease in one week), and lethargy), large abdominal masses, significant skin lesions, and prolapse, according to the guidelines of the Mayo Clinic IACUC to avoid severe pain and distress before death in animals. These symptoms were diagnosed by the expert on-site veterinarian, according to the NIH Body Condition Scoring system (*Burkholder et al., 2012*). Two mice (2% of the total mice) were also euthanized for reasons not related to these health issues; the animals survived until the end of our project, and thus, they were censored in the whole survival analysis. The remaining 66.1% of mice died naturally. A significant difference in the number of euthanized/naturally-dead mice was not observed among *APOE* genotype groups. In the second cohort employed for biochemical analyses, we analyzed the same apoE2-, apoE3- and apoE4-TR mice as we have previously reported (*Shinohara et al., 2016*), except *Apoe*-KO mice were also included. All experiments were conducted in a blinded manner, and all date were included in the analyses without defining outliers nor exclusion criteria.

## Behavioral testing

All tests were performed during the first half of the light cycle. All mice were acclimated to the testing room for 1–2 hr before testing. All behavioral equipment was cleaned with 30% ethanol between each animal. All mice were returned to their home cages and home room after each test.

## Open-field assay

Mice were placed in the center of an open-field area (40 $\times$ 40$\times$30 cm, width x length x height) and allowed to roam freely for 15 min. Side-mounted photobeams raised 7.6 cm above the floor were used to measure rearing while an overhead camera was used to track movement including total distance traveled, average speed, and time spent mobile, with AnyMaze software (Stoelting Co, Wood Dale, IL, USA).

## Elevated plus maze test

As a formal test of anxiety/exploration, the entire maze is elevated 50 cm above the floor and consists of four arms (50 × 10 cm) with two of the arms enclosed with roofless gray walls (35 × 15 cm, L × H). Mice were tested by placing them in the center of the maze facing an open arm, and their behavior was tracked for 10 min using an overhead camera and AnyMaze software.

## Rotarod

Motor coordination was measured using an automated rotarod system (Rotamex-5 Columbus Instruments). The spindle dimensions were 3.0 cm ×9.5 cm and the speed of the rod was set to 4–40 rpm acceleration, increasing 1 rpm every 5 s. The apparatus was equipped with a sensor that automatically stops the timer if the mice fall off the rod. Mice were trained for 4 days in two consecutive trials per day, allowing 10 min of rest between trials.

## Tissue harvest and sample preparation

We harvested mice and prepared samples for biochemical analyses as previously described (*Shinohara et al., 2016*).

## ELISAs and other biochemical assays

Levels of apoE, postsynaptic density 95 (PSD95), glial fibrillary acidic protein (GFAP), CD11b, tumor necrosis factor α (TNFα) and interleukin-1β (IL1β), and cholesterol in the brain were determined, as previously described (*Shinohara et al., 2016*). Levels of C-reactive protein (CRP) were determined using a commercial ELISA kit (R & D Systems). Plasma levels of total cholesterol, HDL-cholesterol and triglycerides were determined using enzymatic assays, according to the manufacturer's instructions (Wako). Plasma levels of non-HDL-cholesterol were determined by subtracting HDL-cholesterol levels from total cholesterol levels in the plasma (*van Deventer et al., 2011*).

## Statistical analysis of animal data

All statistical analyses were performed using JMP Pro software (version 12, SAS Institute Inc). Log-rank tests that were adjusted for sex were used to compare survival between *APOE* genotype groups; p-values<0.0083 were considered as statistically significant after applying a Bonferroni correction for multiple comparisons for the six pair-wise comparisons that were performed. Given evidence of a different survival pattern for apoE3-TR mice over time in comparison to the other groups, specifically with dramatically different pattern after day ~800, we also made pair-wise comparisons of 'early survival' with the apoE3-TR group by censoring all survival times at day 800; p-values<0.0167 were considered as statistically significant after applying a Bonferroni correction for multiple comparisons for the three additional comparisons that were performed.

For behavior experiments and biochemical experiments, comparisons of outcomes across *APOE* genotypes were made (1) by linear regression models adjusting for sex followed by Tukey's HSD test when both genders were included, and (2) one-way ANOVA followed by Tukey-Kramer test when only one gender was included. Comparisons between young and old mice within each *APOE* genotype group were made by linear regression models adjusting for sex followed by pair-wise two-sided Student's t tests. To assess the association between age at death, behavior experiment results, and/or biochemical measurements, the Pearson correlation test was conducted to calculate correlation coefficients (r) and p-values.

Details of statistical tests are also provided in each of the Figure legends. Unless otherwise noted, p-values less than 0.05 were considered significant after appropriate adjustment for multiple comparisons.

## Acknowledgements

General: We thank Ms. Nancy N Diehl, Ms. Sarah E Monsell, Ms. Lilah M Besser, Ms. Merilee A Teylan, and Mr. Zachary Miller for assisting with the analysis of the NACC clinical records, Mr. Michael S Penuliar for assisting with the behavioral experiments, Drs. Mary Jo Ladu and Leon M Tai for providing the apoE ELISA protocol, Dr. Miao Wang for assisting with the lipidomics study, Dr. Douglas C Page, Ms. Theresa G Stile and other Mayo animal facility staff members for caring for the animals,

Olivia N Attrebi for carefully reading this manuscript, and Dr. Yu Yamazaki and Dr. Naoyuki Sato for kindly providing assistance and participating in discussions. The NACC contributors are described in *Supplementary file 1g*.

## Additional information

### Funding

| Funder | Grant reference number | Author |
|---|---|---|
| National Institute on Aging | RF1AG057181 | Guojun Bu |
| National Institute on Aging | R37AG027924 | Guojun Bu |
| National Institute on Aging | R01AG046205 | Guojun Bu |
| National Institute on Aging | RF1AG051504 | Guojun Bu |
| National Institute on Aging | P01NS074969 | Guojun Bu |
| National Institute on Aging | P30AG062677 | Guojun Bu |
| Cure Alzheimer's Fund | | Guojun Bu |
| National Institute on Aging | R21AG052423 | Takahisa Kanekiyo |
| Japan Heart Foundation | | Mitsuru Shinohara |
| Naito Foundation | | Mitsuru Shinohara |
| BrightFocus Foundation | | Mitsuru Shinohara |
| National Center for Geriatrics and Gerontology | | Mitsuru Shinohara |
| Hori Sciences and Arts Foundation | | Mitsuru Shinohara |
| NACC Junior Investigator Award | | Mitsuru Shinohara |

The funders had no role in study design, data collection and interpretation, or the decision to submit the work for publication.

### Author contributions

Mitsuru Shinohara, Conceptualization, Data curation, Formal analysis, Funding acquisition, Investigation, Methodology, Writing - original draft, Project administration, Writing - review and editing; Takahisa Kanekiyo, John D Fryer, Conceptualization, Data curation; Masaya Tachibana, Aishe Kurti, Motoko Shinohara, Yuan Fu, Jing Zhao, Xianlin Han, Data curation; Patrick M Sullivan, G William Rebeck, Resources; Michael G Heckman, Formal analysis, Writing - original draft, Writing - review and editing; Guojun Bu, Conceptualization, Supervision, Funding acquisition, Project administration, Writing - review and editing

### Author ORCIDs

Mitsuru Shinohara (iD) https://orcid.org/0000-0003-3045-7338
John D Fryer (iD) http://orcid.org/0000-0003-3390-2994

### Ethics

Animal experimentation: All cohorts examined in this study were generated from homozygous breeding pairs, group housed without enrichment structures in a specific pathogen-free environment in ventilated cages and used in experiments according to the standards established by the Mayo Clinic Institutional Animal Care and Use Committee (IACUC, Protocol# A58312).

### Decision letter and Author response

Decision letter https://doi.org/10.7554/eLife.62199.sa1
Author response https://doi.org/10.7554/eLife.62199.sa2

# Additional files

## Supplementary files

• Supplementary file 1. Supplementary information of additional analyses etc. (a) Subject characteristics in the overall NACC cohort. The sample median (minimum, maximum) is given for continuous variables. Information was unavailable regarding hypertension (N = 28), transient ischemic attack (N = 116), pacemaker (N = 260), angioplasty/endarterectomy/stent (N = 19), heart attack/cardiac arrest (N = 42), cardiac bypass procedure (N = 9), atrial fibrillation (N = 48), hypercholesterolemia (N = 102), congestive heart failure (N = 25), and stroke (N = 45). (b) Subject characteristics for individuals in the NACC cohort with a neuropathological assessment. The sample median (minimum, maximum) is given for continuous variables. Information was unavailable regarding hypertension (N = 3), transient ischemic attack (N = 24), pacemaker (N = 1), heart attack/cardiac arrest (N = 4), atrial fibrillation (N = 3), hypercholesterolemia (N = 17), congestive heart failure (N = 4), stroke (N = 7),. density of neocortical neuritic plaques CERAD score (N = 12), density of diffuse plaques CERAD score (N = 328), Braak NFT stage (N = 55), presence of vascular pathology (N = 51), and minimal amyloid pathology. (c) Association between *APOE* genotype and survival using the NACC data. HR = hazard ratio; CI = confidence interval; AD = Alzheimer's disease; CV = cardiovascular. HRs, 95% CIs, and p-values result from Cox proportional hazards regression models. [1] These analyses were only performed when considering all subjects. [2] Cognitive status at final visit and AD at final visit were adjusted for only in the model involving all subjects. [3] CV factors included hypertension, transient ischemic attack, pacemaker, angioplasty/endarterectomy/stent, heart attack/cardiac arrest, atrial fibrillation, hypercholesterolemia, congestive heart failure, and stroke. [4] For the neuropathological assessment cohort, all models were additionally adjusted for CERAD diffuse plaque score, CERAD neuritic plaque score, Braak NFT stage, and the presence of vascular pathology. p-values<0.025 are considered as statistically significant after applying a Bonferroni correction for multiple testing for the two primary comparisons of survival that were made (i.e. between *APOE3* and *APOE4* subjects, and between *APOE3* and *APOE2* subjects). (d) Summary of mouse cohorts. For continuous data, values are means ± SD [range]. P-values were calculated using one-way ANOVA (continuous data), the Pearson's chi-square test (categorical value) or log-rank test (median age at death). (e) Effects of *APOE* on 'Dropped activities and interests' and other GDS items in cognitively normal subjects over 60 years old. ORs, 95% CIs, and p-values result from binary logistic regression models for analysis of individual GDS items, and from proportional odds logistic regression models for analysis of the ordinal total GDS score, which is described as the mean (25th percentile, 75th percentile). Information was unavailable for GDS items for a maximum of 23 subjects. (f) Associations between activity levels in the OFA and protein/lipid levels across *APOE* genotype groups at old age. Correlation coefficients (r) and p-values were calculated using the Pearson correlation test. P-values were corrected by Bonferroni test adjusted for the number of protein/lipids analyzed in the entire cohort. CX = cortical, HP = hippocampal. (g) List of NACC contributors.

• Transparent reporting form

## Data availability

All source data files of animal experiments (Figure 2, Figure 4, Supplementary File 1d, and Supplementary File 1f) are included in the manuscript and supporting files. The clinical data are available from NACC (https://www.alz.washington.edu/WEB/researcher_home.html) upon request: distributing any data to a third party, who is not a collaborator or co-authors, is strictly prohibited by NACC.

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
