## [Decision Letter]

**Acceptance summary:**

We believe our readership will be very interested in these interesting new data showing that the association between APOE2 and longer lifespan persists independently of effects on AD status and neuropathology.

**Decision letter after peer review:**

Thank you for submitting your article "APOE2 promotes longevity independent of Alzheimer's disease" for consideration by *eLife*. Your article has been reviewed by three peer reviewers, one of whom is a member of our Board of Reviewing Editors, and the evaluation has been overseen by Huda Zoghbi as the Senior Editor. The reviewers have opted to remain anonymous.

The reviewers have discussed the reviews with one another and the Reviewing Editor has drafted this decision to help you prepare a revised submission.

Summary:

The reviewers were overall positive about your study but have suggested some revisions, which would include additional data, if available.

Essential revisions:

Reviewer 1 asked whether additional memory-based behavioral tasks were performed and can be added, if available. Reviewer 2 asked if it is possible that the increased activity in APOE2 carriers is related to improved cerebrovascular function. For example, Conejero-Goldberg et al., 2014, showed that ECM/integrin-related upregulations, cyclophilin A (CypA) downregulation, and increases in collagen VI and laminin might help in the maintenance of the BBB integrity in APOE2 carriers. Recent studies have shown that the BBB breakdown is associated with cognitive impairment independently of AD amyloid and tau pathology (Nation et al., 2019). Moreover, the BBB breakdown is accelerated in individuals carrying the ε4 allele of the APOE compared to APOE3 homozygotes, and can predict cognitive impairment independently of amyloid and/or tau pathology (Montagne et al., 2020). Although the later study did not include APOE2 carriers gene it is still possible that stabilization of the vascular function, particularly the BBB integrity could contribute to APOE2 protective effects independently of AD pathology. This particularly as the pro-inflammatory CypA-MMP9 pathway is involved in BBB breakdown in APOE4 carriers (Montagne et al., Nature, 2020), and is downregulated in APOE2 carriers as shown by Conejero-Goldberg et al. The authors should discuss these three important papers in relation to their current findings and leave the door open as to whether vasculoprotection could partly explain the protective effect of APOE2.

Reviewer 3 points out that certain associations with plasma lipids and other factors (e.g. Triglycerides and rearing behavior and triglycerides and plasma apoE) may be driven by the fact that ApoE2 KI have very high plasma cholesterol and triglycerides in the plasma and not necessarily by the differences in the lipid profiles. Is there any evidence that correlations like this can be seen only within one apoE genotype or when one controls for apoE genotype?

---

## [Author Response]

Essential revisions:Reviewer 1 asked whether additional memory-based behavioral tasks were performed and can be added, if available.

We appreciate the reviewer 1’s comments on our manuscript. In the first cohort employed for the survival analysis, we did not perform any memory-based behavioral tasks, as these tasks are generally invasive, potentially affecting animal survival. In the second cohort employed for biochemical analyses, we have performed water maze test to assess effects of APOE genotype on memory. That result was already reported in Shinohara et al., Annals of Neurology 79(5):758-774, 2016. The current manuscript used only data which was not reported in the previous paper.

Reviewer 2 asked if it is possible that the increased activity in APOE2 carriers is related to improved cerebrovascular function. For example, Conejero-Goldberg et al., 2014, showed that ECM/integrin-related upregulations, cyclophilin A (CypA) downregulation, and increases in collagen VI and laminin might help in the maintenance of the blood-brain barrier (BBB) integrity in APOE2 carriers. Recent studies have shown that the BBB breakdown is associated with cognitive impairment independently of AD amyloid and tau pathology (Nation et al., 2019). Moreover, the BBB breakdown is accelerated in individuals carrying the ε4 allele of the APOE compared to APOE3 homozygotes, and can predict cognitive impairment independently of amyloid and/or tau pathology (Montagne et al., 2020). Although the later study did not include APOE2 carriers gene it is still possible that stabilization of the vascular function, particularly the BBB integrity could contribute to APOE2 protective effects independently of AD pathology. This particularly as the pro-inflammatory CypA-MMP9 pathway is involved in BBB breakdown in APOE4 carriers (Montagne et al., 2020), and is downregulated in APOE2 carriers as shown by Conejero-Goldberg et al. The authors should discuss these three important papers in relation to their current findings and leave the door open as to whether vasculoprotection could partly explain the protective effect of APOE2.

We appreciate the reviewer 2’s very instructive comments on the potential role of APOE2. We have now included these three important papers and added discussion of necessity to study potential protective effects of APOE2 on vascular function and integrity (Discussion, sixth paragraph).

Reviewer 3 points out that certain associations with plasma lipids and other factors (e.g. Triglycerides and rearing behavior and triglycerides and plasma apoE) may be driven by the fact that ApoE2 KI have very high plasma cholesterol and triglycerides in the plasma and not necessarily by the differences in the lipid profiles. Is there any evidence that correlations like this can be seen only within one apoE genotype or when one controls for apoE genotype?

We appreciate the reviewer 3’s critical and important comments on the necessity to check correlation within each APOE genotype. Indeed, several positive correlations between activity levels and apoE or lipid levels disappeared after stratifying APOE genotype, while correlations between apoE and lipid levels mostly survive even after stratifying APOE genotype. One possibility is that animal behavior scores easily affected by individual variations, or the time lag (~one month) between behavior test and tissue harvesting could contribute additional variability to observe clear correlation within each genotype with limited number of mice. Another possibility is that factors other than apoE and associated lipids might be involved in APOE2-mediated longevity, while the beneficial effects of these apoE-associated lipids (i.e., HDL-cholesterol or triglyceride) on longevity were previously reported as discussed. We have described these points in the Discussion (fifth paragraph).